# Botulinum Toxin Treatment for Cancer-Related Disorders: A Systematic Review

**DOI:** 10.3390/toxins15120689

**Published:** 2023-12-08

**Authors:** Delaram Safarpour, Bahman Jabbari

**Affiliations:** 1Department of Neurology, Oregon Health & Science University, Portland, OR 97239, USA; safarpou@ohsu.edu; 2Department of Neurology, Yale University School of Medicine, New Haven, CT 06510, USA

**Keywords:** botulinum toxin, botulinum neurotoxin, cancer, radiation, pain, gastroparesis, esophageal stricture, parotid gland, parotid fistula, sialocele, gustatory hyperhidrosis, first bite syndrome

## Abstract

This systematic review investigates the effect of botulinum neurotoxin (BoNT) therapy on cancer-related disorders. A major bulk of the literature is focused on BoNT’s effect on pain at the site of surgery or radiation. All 13 published studies on this issue indicated reduction or cessation of pain at these sites after local injection of BoNTs. Twelve studies addressed the effect of BoNT injection into the pylorus (sphincter between the stomach and the first part of the gut) for the prevention of gastroparesis after local resection of esophageal cancer. In eight studies, BoNT injection was superior to no intervention; three studies found no difference between the two approaches. One study compared the result of intra-pyloric BoNT injection with preventive pyloromyotomy (resection of pyloric muscle fibers). Both approaches reduced gastroparesis, but the surgical approach had more serious side effects. BoNT injection was superior to saline injection in the prevention of esophageal stricture after surgery (34% versus 6%, respectively, *p* = 0.02) and produced better results (30% versus 40% stricture) compared to steroid (triamcinolone) injection close to the surgical region. All 12 reported studies on the effect of BoNT injection into the parotid region for the reduction in facial sweating during eating (gustatory hyperhidrosis) found that BoNT injections stopped or significantly reduced facial sweating that developed after parotid gland surgery. Six studies showed that BoNT injection into the parotid region prevented the development of or healed the fistulas that developed after parotid gland resection—parotidectomy gustatory hyperhidrosis (Frey syndrome), post-surgical parotid fistula, and sialocele. Eight studies suggested that BoNT injection into masseter muscle reduced or stopped severe jaw pain after the first bite (first bite syndrome) that may develop as a complication of parotidectomy.

## 1. Introduction

Botulinum neurotoxins (BoNTs) are now major therapeutic agents for a large number of medical disorders based on their ability to inhibit the release of acetylcholine and pain neurotransmitters from presynaptic vesicles [1,2]. They are considered the treatment of choice for several movement disorders, such as cervical dystonia, blepharospasm, and hemifacial spasm [3]. The same anticholinergic mechanism makes BoNTs very useful for the treatment of spasticity, a common symptom associated with conditions such as stroke, cerebral palsy, and brain or spinal cord trauma [4].

Both type A and type B toxins have the potential to inhibit or reduce the action of pain transmitters such as calcitonin gene-related peptide (CGRP), substance P, or glutamate, both in the peripheral and the central nervous system [5,6,7]. OnabotulinumtoxinnA has been approved for the treatment of migraines in the US and European federations since 2010. There is strong evidence that both type A and type B are effective against several other disabling pain disorders [8,9,10,11,12].

Over the past 25 years, BoNT therapy has been studied and used for the treatment of a variety of issues associated with cancer. These include treatment of different types of pain associated with cancer, as well as other cancer-related disorders such as gastroparesis secondary to surgical treatment of local esophageal cancer, post-surgical esophageal stricture, and parotid fistula and gustatory hyperhidrosis that may develop following parotid cancer surgery and radiation. The purpose of this review is to provide up-to-date information regarding the abovementioned potential indications of botulinum toxin therapy for cancer-related disorders. The most important gain from conducting this review will be if any of the published literature has reached the level where it can confidently recommend this treatment for cancer-related disorders or justify conducting phase II or phase III clinical trials in this area.

## 2. Search Design and Search Results

We have searched articles published in Pub Med, Ovid Embrace, and Google Scholar from January 1970 to 1 October 2023, crossing the words cancer (or neoplasm), cancer surgery, radiation with botulinum toxin, or botulinum neurotoxin. Articles not in English were excluded unless the article’s abstract in English provided the minimum information for this review, i.e., number of patients, type of study (double-blind, prospective, retrospective), type and dose of the toxin used, type and location of cancer, location of botulinum toxin injection results, and side effects.

## 3. Results

Of 957 articles found, 47 met the search criteria and underwent final analysis (Figure 1—Prisma). There were four major clinical categories comprising the effect of botulinum neurotoxins (BoNTs) on cancer-associated symptoms: 1—cancer-related pain; 2—gastroparesis caused by esophagectomy for esophageal cancer; 3—esophageal stricture after focal esophagectomy; 4—complications of parotidectomy or parotid radiation (gustatory hyperhidrosis, fistula, sialocele).

## 4. Cancer-Related Pain

Pain related to cancer can be due to direct pressure from cancer upon pain-sensitive structures, remote effect of cancer upon muscles, or post-radiation/post-surgical pain at the region of scarred tissue.

*Direct effect:* A malignant tumor (primary or metastatic) can directly press against adjacent structures and cause pain. There are reports that injection of BoNTs into the tumorous tissue can significantly reduce such pains. The existing literature, however, is limited to two case reports [13,14].

The authors have successfully treated severe jaw pain and locked jaw due to metastasis from lung cancer with BoNT in a patient who had developed the inability to fully open her mouth and severe pain in the area of the right masseter when attempting to open the mouth. The problem gradually reached a point where she refrained from eating. Medications, including oxycodone and fentanyl, provided moderate and temporary pain relief but did not alleviate her trismus. A magnetic resonance imaging (MRI) of the jaws demonstrated enlargement of the right masseter muscle due to involvement by metastatic carcinoma (Figure 2). Injection of 50 units of onabotulinumtoxinA into the right masseter and 20 units into the right temporalis muscles decreased the right masseter pain substantially and improved the jaw opening for a period of 6 weeks. Subsequent injections of a larger dose of onaA into the right masseter (70 units) with an additional injection into the left masseter (30 units) enabled her to eat and improved her quality of life (pain relief, less eating difficulty) over the next 18 months before her demise from complications of cancer.

Another report [14] describes two patients suffering from severe local pain due to the involvement of the psoas muscle by metastatic carcinoma and pressure by adjacent enlarged lymph nodes. Injection of 75 to 100 units of OnaA within two weeks resulted in a marked improvement in pain that lasted for 12 weeks. The main mechanism of pain relief in our patient and the other two reported cases [14] is probably similar (i.e., reducing pressure upon the adjacent nerve(s) due to relaxation of nearby muscles). However, the inhibitory effect of the botulinum toxin upon pain transmitters, as described earlier, can be another contributory factor.

*Remote Effect:* Malignant tumors, when located inside the brain or spinal cord through affecting motor or sensory fibers, can cause remote effects. These effects are usually in the form of muscle spasms. Two case reports on the remote effects of such cancers are described below. The literature on the remote analgesic effect of botulinum toxin on cancer pain is also limited to two case reports.

The authors have used BoNT for the treatment of disabling, deep neck and shoulder pain associated with extensive ponto-medullary astrocytoma (Figure 3) in a young patient who suffered from spasms of the neck and shoulder muscles. The patient had failed treatments with non-steroidal anti-inflammatory analgesics and tizanidine, and treatment with opioids had minimal analgesic effect. The following muscles were injected with onabotulinum toxinA: left and right splenius capitis (40 units each), left and right trapezius (40 units each), left and right levator scapulae (40 units each), and left and right sternocleidomastoid (20 units each). The total dose was 280 units. The patient reported significant pain relief following these injections. His initial pain score on the Visual Analogue Scale (VAS) was between 8 and 9. It dropped to 3 to 4 a week following injections. The analgesic effect of the injected toxin lasted between 2.5 and 3 months. Injections were repeated every three months for two years until the patient passed away from complications of cancer.

Nam and co-workers [15] have reported a 62-year-old male with intracranial chondrosarcoma who suffered from neuropathic pain (burning) in a large area involving the back of the left thigh. Injection of 100 units of onabotulinumtoxinA distributed into 16 regions of the posterior thigh reduced the pain significantly (the pain level on the Visual Analogue Scale dropped from 6 to 2). The mechanism of pain relief in our case and the one reported by Nam and co-worker seems to be different. While in our case, the pain relief seems to be related to toxin-induced relaxation of painful neck muscles, the pain relief in the patient of other reports most likely resulted from the inhibitory effect of onabotulinumtoxinA upon pain transmitters.

## 5. Pain Following Surgery and/or Radiation

Pain at the site of surgery and radiation is common among cancer patients. Most reported patients have had throat, tongue, and neck cancer. Table 1 represents the search results up to 1 October 2023.

These positive results are supported by several individual case reports, including one in a patient with Raynaud syndrome secondary to lung cancer in whom severe palm pain improved significantly (3 grades in VAS) after injection of onabotulinumtoxinA at multiple points into the palm [30] (Figure 4).

## 6. Botulinum Toxin Therapy after Esophagectomy and for Gastroparesis

Esophagectomy may lead to gastroparesis and esophageal stricture. Delayed gastric emptying (DGE) is believed to occur in up to 50% of the patients after esophagectomy for cancer [31]. Delayed gastric emptying following esophagectomy can increase the risk of aspiration, leading to prolonged hospital stays and decreasing the patient’s satisfaction after surgery. The searched literature suggests the utility of local injection of botulinum toxins in these disorders (Table 2).

Table 2 shows 12 studies, prospective and retrospective but no blinded investigation, describing the effect of botulinum toxin injections on DGE or preventing the development of DGE after esophagectomy. In seven studies, BoNT was injected during and in four studies after esophagectomy. One study evaluated the effect of combined BoNT injection and ballooning on post-esophagectomy DGE. In three of these studies, the dose of BoNT is not mentioned [38,39,41], and in one of the three [39], the authors did not mention which type of A toxin was used. Since the efficacy of the BoNT is highly dose-dependent, the results of these three studies cannot be compared with the rest; hence, they are not included in the final analysis. Unfortunately, these three studies are included in a recently published meta-analysis of this subject, making the authors’ conclusions debatable [44]. Among the remaining studies, six found that intra or postoperative injection of BoNTs into pyrolus led to the development of DGE in fewer patients, while three did not. In one study [43], intrapyloric injection of onabotulinumtoxinA combined with balloon dilatation in 85% of the patients improved the severity of DGE by 50% or more.

## 7. Esophageal Stricture (ES)

Esophageal stricture is common after esophagectomy and, depending on the esophageal pathology, affects 4–46% of the patients after esophageal surgery [45]. Dysphagia is a common symptom of ES. Severe cases may require tube feeding. Milder cases of ES are treated with balloon dilation, which may be repeated if necessary. More severe cases may require intralesional steroid injection or reconstructive surgery [46].

Wen et al. [47], in a double-blind, placebo-controlled study, investigated the effect of BoNT injections in 72 patients affected by ES caused by esophagectomy for squamous cell carcinoma with esophageal stent (ESD). Patients were divided into two groups: one group received an injection of BoNT-A (Lanzhou Institute of Biological Products, Lanzhou, China) immediately after dissection and ESD, while the other group served as a control and received no BoNT-A. One hundred units of the BoNT-A were divided into 10-unit portions and injected into ten points around the area of resection. A follow-up endoscopy defined the stricture formation when the esophageal lumen was less than 9.8 mm, not allowing the passage of a standard endoscope through the stenotic area. The secondary outcome measure was the number of esophageal balloon dilations required after surgery. The BoNT group demonstrated less development of stenosis compared to the control group: 6.1% versus 32.4% (*p* = 0.02). The BoNT group also required fewer esophageal dilations than the control group (*p* = 0.002). Dysphagia grading using the Mellow–Pinkas dysphagia score was also lower in the BoNT-treated group.

In another study, Zhou et al. [48] prospectively compared the preventive action of BoNT injection against development of strictures with steroid (triamcinolone) and no intervention (control group) in 80 patients who underwent submucosal resection of esophageal cancer. The BoNT group received 100 units of BoNT-A (Lanzhou Institute of Biological Products, Lanzhou, China) injected along the junction of the defect and normal tissue at 10 points (10 units/point). The total dose of triamcinolone was 40mg and injected deeper, close to the edge of the ulcer. The proportion of patients developing stricture (primary outcome) was 30% in the BoNT group, 40.9% in the triamcinolone group, and 82.4% in the control group (*p* < 0.001 and *p* < 004).

In a recent retrospective study of 204 patients [49], the authors compared the effect of oral statins, BoNT injection, oral and topical steroids, and non-intervention with each other in reducing the risk of stricture development after esophagectomy. The authors found that the effect of statins and BoNT injections in preventing the development of strictures were comparable with each other, and the stricture occurrence rates in the statins and BoNT group were significantly lower than both the steroid treatment and the non-intervention approach (Table 3).

The positive results of the abovementioned studies in humans are supported by several animal studies that show injection of BoNTs close to the site of esophagectomy can reduce the development of esophageal strictures [50,51].

## 8. Parotidectomy and Parotid Radiation

Parotidectomy for the removal of a cancerous parotid gland or radiation of an affected parotid gland can lead to several medical disorders. Gustatory hyperhidrosis (GH-facial sweating during eating) or Frey syndrome (named after a Polish pathologist) is encountered in 20–60% of the patients after parotidectomy [52]. It impairs the quality of life in 15–30% of the patients. Parotid gland radiation can lead to fistula and sialocele formation. First bite syndrome (severe jaw pain at first bite), though uncommon, is another side effect of parotidectomy. Our literature search found 18 articles regarding BoNT therapy for GH, fistula, and sialocele formation (Table 4).

The normal flow of saliva interferes with the healing of fistulas caused by surgery or radiation. Injection of BoNTs into parotid glands reduces the flow of saliva through the inhibitory effect of the toxin-blocking acetylcholine release from parasympathetic fibers that innervate the salivary glands. It has been shown that a significant decrease in the flow of saliva occurs approximately 4 days after intra-parotid BoNT injection [61].

## 9. First Bite Syndrome

First bite syndrome (FBS) is an uncommon complication of parotidectomy caused by damage to the sympathetic branches innervating the parotid gland and the development of denervation hypersensitivity from unopposed parasympathetic stimulation of the salivary gland’s myoepithelial cells causing pain at the onset of gustatory salivation [71]. Sheik et al. [72] recently reviewed the literature (eight manuscripts, 22 patients) on BoNT therapy in first bite syndrome following parotidectomy; there were no blinded studies, and most reports were retrospective observations. Symptom improvement occurred in all eight patients injected with 40 units of onabotulinumtoxinA. In total, 7 of the 22 patients (38.1%) had complete resolution of FBS after BoNT injection. No side effects were reported.

## 10. Discussion

Botulinum neurotoxin’s molecule contains a sophisticated machinery that allows the toxin to exert its function within the nerve cell after intramuscular or subcutaneous injection. Each molecule of the toxin consists of a light chain and a heavy chain bound together by a disulfide bond. The heavy chain of the toxin attaches the toxin to the nerve cell surface and initiates the process of entry into the cell (receptor binding and internalization), while the light chain (a protease) exerts the toxin’s function inside the cell by attaching itself to and deactivating specific intracellular proteins (SNARE proteins). SNARE proteins are essential for the rupture of presynaptic vesicles and the release of neurotransmitters cleft (acetylcholine in case of neuromuscular junction) into the synaptic [73]. Of eight recognized toxin serotypes (A–G), types A and B are currently in clinical use due to their long duration of action. Most of the literature on the role of BoNTs in the treatment of cancer-related disorders relates to the action of BoNTs upon post-surgical or post-radiation pain (Table 1). The pain experienced in these conditions is often neuropathic in type, i.e., having a searing or burning quality. In a recent review of the literature, Matak and co-workers [74] presented convincing evidence that BoNTs can reduce the function of pain transmitters (substance P, calcitonin gene-related peptide, glutamate, and others) at cellular, peripheral nerve, and central nervous system levels.

Adding BoNT to the cell culture (trigeminal or dorsal root ganglia neurons) blocks the KCL-evoked release of two major pain neurotransmitters, namely calcitonin gene-related peptide (CGRP) and substance P (SP) [75,76]. A similar effect has been observed in ex-vivo bladder preparations [77,78]. Exposure to BoNTs reduces glutamate release from dorsal horn neurons of the spinal cord [79].

At the peripheral nerve level, injection of onabotulinumtoxinA into the rat’s paw, prior to injection of formalin, reduced the release of glutamate from peripheral nerves into the tissue and alleviated formalin-induced pain [80]. In rat bladders exposed to hydrochloric acid, the release of SP and CGRP was partially inhibited after exposure of the bladder tissue to BoNT-A [78]. BoNT-A decreases the function of sodium channels substantially [81]. Sodium channels are abundantly present on the peripheral nerves and on the central sensory fibers. They are believed to be essential in conducting peripheral sensations (including pain) in the brain [82]. Double-blind, placebo-controlled studies have shown that subcutaneous injection of BoNTs (A or B) can significantly reduce several neuropathic types of pain in humans [7,9,10,11,12].

There is now ample evidence that the analgesic effect of BoNT is also exerted at the central level [7]. Several studies have shown that in animals with experimental bilateral painful peripheral neuropathy, unilateral injection of BoNT into a limb improves the neuropathic pain bilaterally [83,84,85]. In another pain model, when injected into the spinal canal, botulinum toxin was more effective against pain and in lower doses compared to peripheral injection [86]. In cancer patients, when pain appears to have a major muscular component (contraction, spasm), the pain relief after BoNT injection may be all or partly due to inhibition of acetylcholine release leading to muscle relaxation (example: case 2, Figure 3). Inhibition of acetylcholine release also explains the prevention of post-parotidectomy fistula and sialocele development, as well as the improvement in gustatory hyperhidrosis after parotid surgery. Emerging data indicate that botulinum toxin injections may interfere with cancer development, an important subject that was recently reviewed by Grenda and co-workers [87]. In in vitro studies, several authors assessed the effect of BoNT application on different cancer cell lines. Exposure to BoNTs slowed the progression and development of breast, prostate, colon cancer cells, and neuroblastoma cells through different molecular mechanisms [88,89,90,91,92]. In in vivo studies, mostly conducted in mice, investigators have shown that exposure to BoNTs slowed down the progression and development of glioblastoma, prostate, fibrosarcoma, and pancreatic cancers and promoted apoptosis in some of these cancers [92,93,94,95].

## 11. Conclusions

The inhibitory effect of BoNTs on the synaptic release of acetylcholine and pain neuromediators makes these agents useful for the treatment of several clinical disorders associated with cancer. These consist of pain associated with cancer, treatment of gastroparesis after esophagectomy, prevention of esophageal stricture, and prevention and treatment of fistula and sialocele following parotidectomy. The data on pain associated with cancer overwhelmingly support the analgesic effect of botulinum toxin injections for cancer-associated pain. The data on gastroparesis is mixed, although a majority of the publications suggest its effectiveness for this indication (Table 2). Results of three studies, including a double-blind, placebo-controlled study, attest to the effectiveness of local BoNT injections in the prevention of esophageal stricture development after esophagectomy. Data regarding the prevention or healing of fistula and sialocele is also promising and suggests the effectiveness of BoNTs in promoting the healing of these lesions via the reduction in saliva secretion. Confirmation of the positive effects of these observations awaits the results of well-designed, controlled, and blinded studies, preferably in a large group of patients. The data from basic scientists on the effect of BoNTs slowing showing growth and development of different cancer cells both in vitro and in vivo is important and promising and deserves further evaluation.

## Figures and Tables

**Figure 1 toxins-15-00689-f001:**
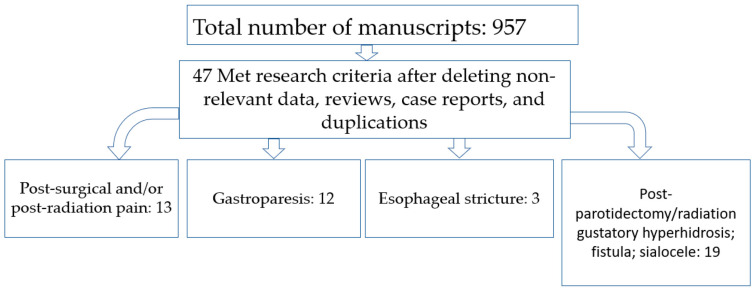
Total number of articles found as well as numbers in different clinical categories.

**Figure 2 toxins-15-00689-f002:**
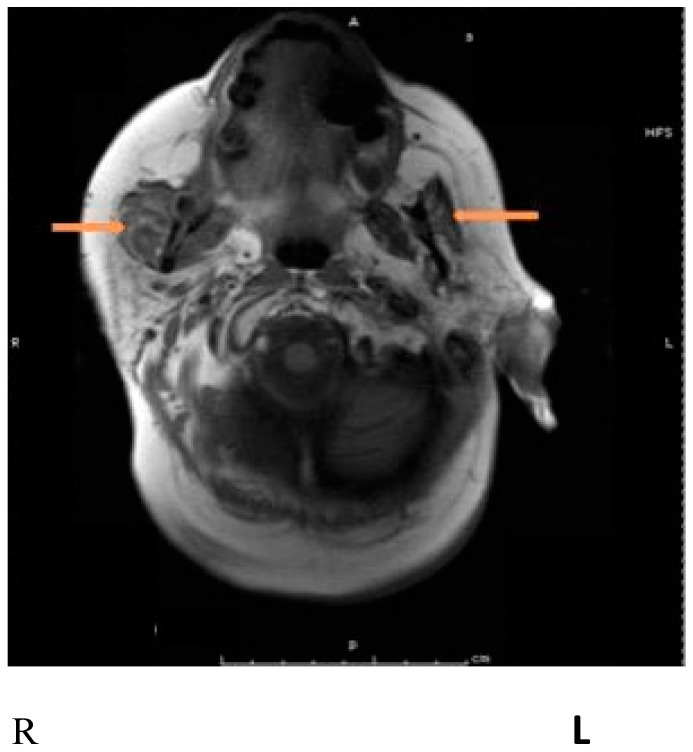
Magnetic resonance imaging (MRI) showing an enlarged masseter on the right side, probably due to metastatic involvement. Arrows pointing to the masseter muscles bilaterally. From Jabbari B. Botulinum toxin treatment of pain disorders. 2nd edition. Courtesy of publisher. Springer. Berlin, Germany [13].

**Figure 3 toxins-15-00689-f003:**
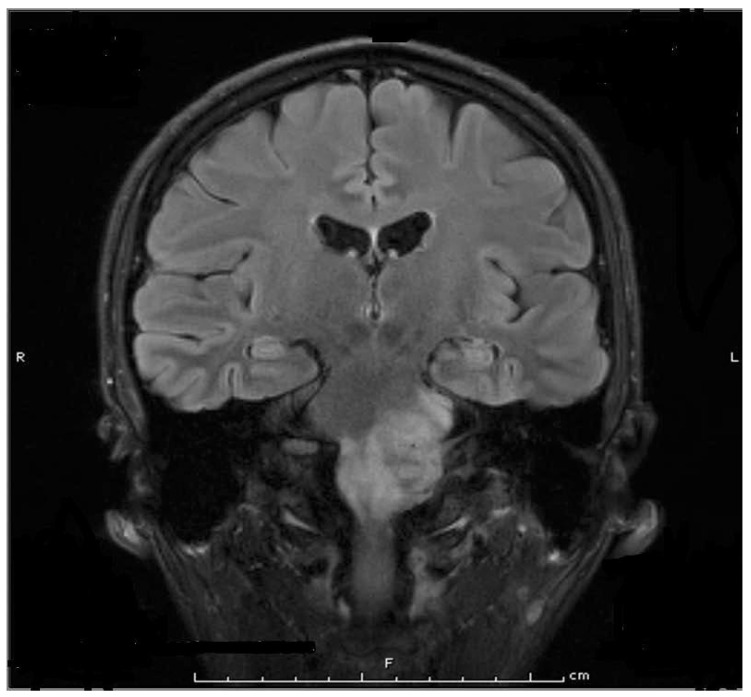
Magnetic resonance scan (MRI) shows a large ponto-medullary tumor. From Jabbari B. Botulinum toxin treatment of pain disorders. 2nd edition. Courtesy of publisher. Springer, Berlin Germany [13].

**Figure 4 toxins-15-00689-f004:**
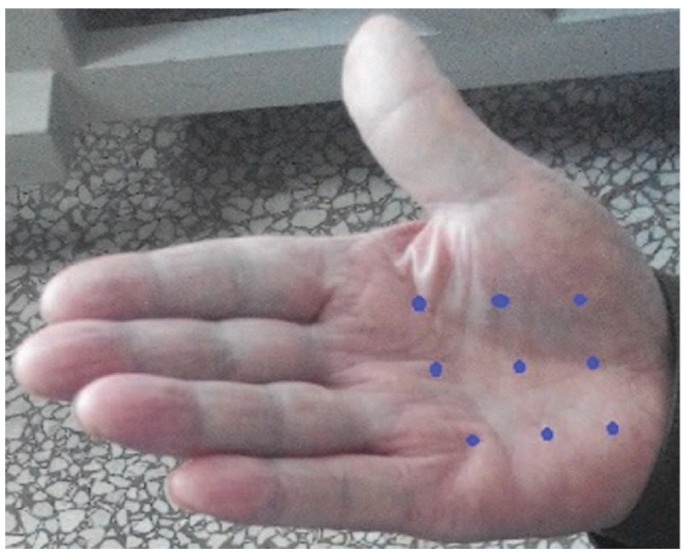
The blue dots point to the site of BoNT-A injection for treatment of severe palm pain in Raynaud syndrome. From Wang et al., 2016 [30]—reproduced under the Creative Commons Attribution License 4.0. Courtesy of the publisher (Wolters-Kluwer, Amsterdam, Holland).

**Table 1 toxins-15-00689-t001:** Published studies on the effect of BoNT on local pain resulting from radiation and/or surgery. Revised and updated from Mittal and Jabbari [16].

Authors, Date	#Pts Study Type	Toxin	Dose Units	Treatment	Type of Cancer	Primary Outcome	Result
Van Deale et al., 2002 [17]	6, Retro	onaA	20–25	Radiation or Chemotherapy	Head and neck	Pain (VAS)	Complete pain relief in 4 of 6 Pts; Significant improvement in quality of life using SF36, EQ-5D scales
Vasan et al., 2004 [18]	16, Pros	aboA	100–320	Surgery	Head and neck	Pain (VAS-days 3 and 4 wks), Global Quality of Life (GQL)	Significant pain reduction (*p* = 0.05) Quality of life improved (*p* = 0.7)
Wittekind et al., 2006 [19]	23, Pros	OnaA	60–120; 160–240	Radiation, Surgery	Head and neck	Pain (VAS) At 28 wks	Significant reduction in pain at lower dose (<0.05)
Hartl et al., 2008 [20]	19, Pros	onaA aboA	50 250	Chemotherapy radiation	Head and neck	Pain: (VAS) Function: At 4 wks	Both improved Pain (*p* =0.02) Function (*p* = 0.04)
Stublefied et al., 2008 [21]	23, Retro	onaA	25–200	Radiation Surgery	Head, neck, breast	Pain (VAS)	Pain Improved in 85% of Pts
Chuang et al., 2008 [22]	8, Retro	onaA	100	Radiation	Bladder Prostate	Pain (VAS)	Pain average dropped from level 8 to 2
Voung et al., 2010 [23]	15, Can 20 Con Pros	onaA	100 unit	Radiation	Pelvic cancer	Pain(VAS)	Significant difference from control (*p* < 0.02)
Mittal et al., 2012 [24]	7, Retro	onaA	100	Radiation Surgery	Head, neck, breast	Pain (VAS), PGIC at 4 weeks	VAS 6 of 7 improved: *p* < 0.05 PGIC: 6 of 7, very satisfied QoL; 6 of 7 improved (*p* < 0.05)
Bach et al., 2012 [25]	9, Pros	onaA	100–400	Radiation and Surgery	Head and neck	Pain (VAS) and FDSNP at 4 weeks	Both pain and FDSNP improved (*p* < 0.01)
Rostami et al., 2014 [26]	12, Pros	incoA	100	Radiation and Surgery	Head, neck, breast	Pain (VAS) and PGIC at week 6	Both VAS (*p* < 0.05) PGIC: very satisfied QoL in 38% (*p* < 0.05)
Degroef et al., 2018 [27]	50, Db-pc	0naA	100	During surgery	Breast cancer	Pain (VAS) at 3 and 6 months	At 6 months, 40% in the intervention group and 60% in control group still had pectoral pain (*p* > 0.05)
Mailly et al., 2019 [28]	16, Retro	incoA aboA	20– 40	Radiation and surgery	Head and neck	Pain (VAS)	VAS improved *p* < 0.01 in 11 pts
Dang et al., 2019 [29]	3, Retro	onaA	20–60	surgery	Base of skull schwannoma	Pain at the site of skull incision	Headaches subsided

onaA: onabotulinumtoxinA; aboA: abobotulinumtoxinA; incoA: incobotulinumtoxinA; VAS: Visual Analogue scale (0–10); PGIC: patient global impression of change FDSNP: functional disability scale for neck pain; Pros = prospective; Retro = retrospective; Db-pc = double-blind, placebo-controlled; QoL = quality of life. #Pts = number of patients. Wks = weeks.

**Table 2 toxins-15-00689-t002:** Botulinum toxin treatment in cancer-related esophagectomy and gastroparesis (excluding single case reports).

Authors and Date	Number of Patients, Study Type	Toxin	Dose in Units, Injected Site	Procedure	Pathology	Results
Kent et al., 2007 [32]	15 12 injected during esophagectomy 3 shortly after surgery; Retro	onaA	200 (Anterior pylorus at 4 points)	During esophagectomy	Not specified	No patient injected with Botox during esophagectomy developed DGE.
Cerfolio et al., 2009 [33]	150, No intervention vs pyloroplasty vs. BoNT-A Retro	OnaA	100 (Divided into 4 injections into pylorus)	During esophagectomy	Adeno and squamous cell carcinoma	Incidence of DGE in no intervention and pyloroplasty: 96%; in OnaA: 56% (*p* = 0.05)
Martin et al., 2009 [34]	45, Intrapyloric injection of Botox during pyloroplasty; Pro	onaA	200 (Into anterior wall of pyrolus)	After esophagectomy	Not specified	96% of patients showed no evidence of DGE (3 months or more)
Bagheri et al., 2013 [35]	60, Pyloroplasty vs. Botox injection; Pros	onaA	100 (Lower section of pylorus)	After esophagectomy	Thoracic, gastric, esophageal cancers	Incidence of DGE in onaA injected group 10% vs. 26% in pyloroplasty group
Antonoff et al., 2014 [36]	293, Pyloroplasty vs. Botox with dilation; Retro	onaA	200 (Different parts of pyrolus)	After esophagectomy	Different cancers	Both interventions were superior to no intervention in preventing DGE. No difference between the two interventions, but pyloroplasty group had two serious side effects,
Eldaif et al., 2014 [37]	322: 86% with esophageal cancer, Pyloroplasty vs. pylomyotomy vs. onaA; Retro	onaA	100 (Divided into 4 separate pyloric Injections)	After esophagectomy Evaluated for DGE between 5–7 postoperative day	Not specified Majority had preoperative radiation	Botox injection decreased operative time but did not change the incidence of DGE
Fuchs et al., 2016 [38]	41, BoNT-A injection vs. no intervention Retro	onaA	200 (Divided into 4 injections into pylorus	During esophagectomy	Adeno and squamous cell carsinomas	DGE Botox: 0 Non-intervention: 8 *p* < 0.05
Stewart et al., 2017 [39]	71, Intraoperative injection of BoNT-A compared with no intervention Retro	onaA	Not mentioned	During esophagectomy	Cancer Type not specified	No difference between onaA injected and no intervention group in duration of jejunostomy tube use
Giugliano et al., 2017 [40]	146, BoNT-A injection compared with no intervention; Retro	BoNT-A	Not mentioned	During esophagectomy	Cancer, 91.8% adenocarcinoma	No difference between BoNT group and no intervention
Marchese et al., 2019 [41]	90, Pyloroplasty vs. onaA injection vs. no therapy; Pros	onaA	200 (Divided into 4 injections pylorus)	During esophagectomy	Not mentioned	Incidence of DGE was the same in all three groups
Tham et al., 2019 [42]	228, Botulinum toxin injection versus no intervention Retro	onaA	Not mentioned	During esophagectomy	Most adenocarcinoma	No difference between Botox injection and no intervention
Bhutani et al., 2022 [43]	21 with DGE after esophagostomy treated with BoNT injection and ballooning	onaA	100 units (Divided into 4 injections into pylorus)	Treating post–esophagectomy DGE	Esophageal cancer	85% reported improvement by more than 50% of DGE symptoms

**Table 3 toxins-15-00689-t003:** Reports of botulinum toxin treatment of esophageal stricture esophagectomy (Search results up to 1 October 2023).

Authors	Design	Number of Patients	Clinical Problem	Injection Site	Toxin and Dose in Units	Result
Wen et al., 2016 [47]	Pro	67	Prevention of ES after esophageal submucosal dissection	Esophagus	Ona-A 100	Decrease in ES, decrease in esophageal dilation
Zhou et al., 2021 [48]	Pro	78	Prevention of ES after esophageal submucosal dissection	Esophagus	Ona-A 100	Decrease in ES, particularly in patients who had entire circumference mucosal defect.
Wang et al., 2023 [49]	Retro	204	Prevention of ES after esophageal submucosal dissection	Esophagus	Ona-A 100	Decrease in ES in patients who received statins or BoNT injection

**Table 4 toxins-15-00689-t004:** Reports on botulinum toxin treatment of gustatory hyperhidrosis after parotidectomy and post-parotidectomy sialorrhea, prevention of fistula and sialocele formation after parotid radiation (Search results up to 1 October 2023).

Authors	Design	Number of Patients	Clinical Problem	Injection Site	Toxin and Dose in Units	Result
Bjerkhoel et al., 1997 [53]	Pro	15	GH after parotidectomy	Face	OnaA 17–62.5	Total cessation of facial sweating in 13 patients
Laccourreye et al., 1998 [54]	Pro	14	GH after parotidectomy	Face	OnaA 25–88	All showed total cessation of sweating
Von Lindern et al., 2000 [55]	Retro	7	GH after parotidectomy	Face	OnaA	Sweating stopped after BoNT injection
Cavalot et al., 2000 [56]	Pro	40	GH after parotidectomy	Face	OnaA, 2.5/cm	100% response in severe group, 72% response in moderate group
Vargas et al., 2000 [57]	Pro	4	Post-parotidectomy sialocele pain	Parotid gland	OnaA, 30–50	Total resolution within 4 weeks in all patients
Kuttner et al., 2001 [58]	Retro	8	GH after parotidectomy	Face	BoNT-A, 0.5/cm	Stopped facial sweating within one week
Eckardt et al., 2003 [59]	Retro	33	GH after parotidectomy	Face	OnaA, 16–80	Facial sweating disappeared within a week after injections
Nolte et al., 2004 [60]	Pro	20	Gustatory sweating after parotidectomy	Facial	OnaA, 3/cm	Total cessation of sweating for 12 months
Marchese-Ragona et al., 2006 [61]	Retro	3	Post-parotidectomy fistula	Parotid gland	OnaA, 15–20	Complete healing of fistula with follow-ups of 12, 18, and 14 months
Pomprasit et al., 2007 [62]	Pro	9	GH after Parotidectomy	Face	OnaA, 10.6	Sweating stopped in 5 pts and was reduced in 4 pts
Machese et al., 2008 [63]	Retro	8	Head and neck cancer: 6 sialorrhea; 1 1 fistula; 1sialocele	Parotid gland	AboA, 100	Fistulas healed. Sialorrhea stopped
Martos Dias et al., 2008 [64]	Retro	10	GH after parotidectomy	Face	OnaA, 38	Sweating stopped
Hatrl et al., 2008 [20]	Retro	7	GH after parotidectomy	Face	OnaA 50 Abo-A 250	Sweating and quality of life improved
Cantarella and Barbieri, 2010 [65]	Retro	7	GH after parotidectomy	Face	RimaB, 2200	Cessation of sweating in 6 of 7 patients 4 weeks after injection
Laskawi et al., 2013 [66]	Retro	10	Post-parotidectomy fistula	Parotid gland	OnaA, 30–50	Treated within 6 weeks of surgery: fistulas healed in 9 of 10 patients
Steffen et al., 2014 [67]	Rro	25	Head and neck cancer: FHS: (19), Fistula (6)	Parotid gland	OnaA and incoA, Parotid: 30 UM: 20	FHS: 11 of 19 improved Fistula: 4 of 6 healed
Melville et al., 2016 [68]	Pro	3	Buccal squamous cell carcinoma; parotid sialocele and fistula	Parotid gland	onaA, 50–70	In all three, fistula and sialocele healed, and serous drainage stopped
Marchese et al., 2022 [69]	Retro	12 All had CT and RT	Cancer of larynx and pharynx, sialocele/fistula	Parotid gland	onaA, 80 into each gland	54% of the patients had closure of fistula within days
Mueller et al., 2022 [70]	Pro	10	Prostate cancer Post Ac-PSMA therapy	face	IncoA, 6 injections, 30 u per injection point	Those injected by BoNT showed a mean of 29% gland destruction after two cycles of Ac-PSMA treatment compared 60–70% seen in those who did not receive BoNT injection

Retro: Retrospective; Pro: Prospective; onaA: OnabotulinumtoxinA; incoA: IncobotulinumtoxinA; aboA: AbobotulinumtoxinA; GH: Gustatory hyperhidrosis; FHS: Functional hypersalivation; Ac-PSMA: Actinium-225-PSMA; CT: chemotherapy; RT: radiotherapy.

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
