# Peer review of "Botulinum Toxin Treatment for Cancer-Related Disorders: A Systematic Review"

_toxins, 2023, doi:10.3390/toxins15120689_

Round 1

Reviewer 1 Report

Comments and Suggestions for Authors

The authors report a systematic literature review of use of botulinum toxin for cancer related disorders. Disorders specifically called-out for review are: pain, reduced movement of food through the esophagus and stomach, excessive sweating, and excessive saliva production. It is valuable to review the literature on these topics. The authors could improve the review by describing more completely which of these uses of botulinum neurotoxin have been taken up widely, by many different groups working independently in different centres, and which are used only by a few individuals and have not yet been taken up more widely by the community. What are the common observations made across multiple different groups for the widely used examples, and are there any inconsistencies between in reported outcomes? What are the most important insights gained from having conducted this literature review?

Major comments:

(1)    The abstract and introduction should set out more fully what questions led the authors to undertake this literature review and summarise the most important insights gained from having done it.

(2)    The example of direct effect cancer pain, caused by pressure from a tumor pressing against adjacent structures (pages 2 – 3, lines 60 – 84), seems out of place in a systematic literature review because it is an example from the author’s personal experience and appears to be one of only two such citations identified in the search results.

(3)    The example of remote effect cancer pain (pages 3 – 4, lines 87 – 102) seems out of place in a systematic literature review because it is an example from the author’s personal experience. Similar to comment two above, the authors did not need to perform a literature review to identify this case from their personal experience. Please add more discussion putting these cases from the author’s own experience into context alongside other published reports and highlighting what are the most common findings consistently reported by a range of different investigators working independently across different centres.

Minor comments:

(1)    The abstract is difficult to understand because it uses only medical terms to describe the cancer related disorders being reviewed. A plain English summary afterwards would be helpful for non-medical readers.

(2)    The manuscript contains several formatting errors. Please proof-read and correct before publication.

Comments on the Quality of English Language

English language is fine. The manuscript contains several formatting errors which should be corrected.

Author Response

The authors report a systematic literature review of use of botulinum toxin for cancer related disorders. Disorders specifically called-out for review are: pain, reduced movement of food through the esophagus and stomach, excessive sweating, and excessive saliva production. It is valuable to review the literature on these topics. The authors could improve the review by describing more completely which of these uses of botulinum neurotoxin have been taken up widely, by many different groups working independently in different centres, and which are used only by a few individuals and have not yet been taken up more widely by the community. What are the common observations made across multiple different groups for the widely used examples, and are there any inconsistencies between in reported outcomes? What are the most important insights gained from having conducted this literature review?

Answer: Following your advice, two sentences are added to the end of introduction to answer some of these questions. (lines 63-68 of marked manuscript) More is discussed in the newly added section of the manuscript (Discussion).

Major comments:

  • The abstract and introduction should set out more fully what questions led the authors to undertake this literature review and summarise the most important insights gained from having done it.

Answer- please see the revised abstract and the last two  sentences of introduction. Lines 63-68of marked manuscript

  • The example of direct effect cancer pain, caused by pressure from a tumor pressing against adjacent structures (pages 2 – 3, lines 60 – 84), seems out of place in a systematic literature review because it is an example from the author’s personal experience and appears to be one of only two such citations identified in the search results.

Answer- Following your suggestion this section is revised (pages 3 and 4 of the manuscript)

  • The example of remote effect cancer pain (pages 3 – 4, lines 87 – 102) seems out of place in a systematic literature review because it is an example from the author’s personal experience. Similar to comment two above, the authors did not need to perform a literature review to identify this case from their personal experience. Please add more discussion putting these cases from the author’s own experience into context alongside other published reports and highlighting what are the most common findings consistently reported by a range of different investigators working independently across different centres.

Answer: Following your suggestion this section is revised (page 3 and 4 of the manuscript)

Minor comments:

  • The abstract is difficult to understand because it uses only medical terms to describe the cancer related disorders being reviewed. A plain English summary afterwards would be helpful for non-medical readers.

Answer: The abstract is revised, expanded and written in a simpler language.

  • The manuscript contains several formatting errors. Please proof-read and correct before publication.

Answer: formatting errors are corrected.

We thank you for your constructive suggestions.

Reviewer 2 Report

Comments and Suggestions for Authors

I have read the manuscript entitled Botulinum Toxin Treatment for Cancer Related Disorders: A Systematic Review submitted for publication to Toxins.

The manuscript describes a systematic evaluation of the literature regarding the use of botulinum toxin for the treatment of cancer-related complications: pain (post-surgical, post-radiation, direct tumor or pressure from metastatic invasion); gastric or esophageal dysfunctions; post paroidectomy complications. The text is of interest to clinicians who are faced with the complications described. A total of 957 studies are cited (it is not clear what time period they include. Maybe the 25 years mentioned in the text in line 28?) summarised in clear tables. However, in my opinion, the manuscript should be addressed not only to a smaller audience of clinicians but also to a wider audience of Toxins readers, including researchers who are not really clinicians. The aim is also to stimulate the interest of these readers and possibly the interactions between basic research and clinical research.

The manuscript sent lacks a description (even a brief one, but precise and clear) of the possible mechanisms through which the effect of the toxin on the pathologies described can be explained. Some of them are obviously known, but a description of them with respect to the specific pathophysiological picture described can be useful to better frame the problem and the way to solve it. In other cases, the possible effect of the toxin from a physiological point of view is less clear. By way of example, recently, the same journal Toxins published a manuscript entitled "Mechanisms of Botulinum Toxin Type A Action on Pain" which I do not see cited, where an accurate analysis of the possible physiological mechanisms of the effect of botulinum toxin on nociceptive sensation was made.

I therefore suggest adding a part to the manuscript dedicated to the molecular and physiological mechanisms through which the toxin would act in the solution of the side effects described in the manuscript.

Therefore I think that this manuscript needs  major revision before it can be acceptable for publication in this Journal

Author Response

The manuscript describes a systematic evaluation of the literature regarding the use of botulinum toxin for the treatment of cancer-related complications: pain (post-surgical, post-radiation, direct tumor or pressure from metastatic invasion); gastric or esophageal dysfunctions; post parotidectomy complications. The text is of interest to clinicians who are faced with the complications described. A total of 957 studies are cited (it is not clear what time period they include. Maybe the 25 years mentioned in the text in line 28?) summarized in clear tables. However, in my opinion, the manuscript should be addressed not only to a smaller audience of clinicians but also to a wider audience of Toxins readers, including researchers who are not really clinicians. The aim is also to stimulate the interest of these readers and possibly the interactions between basic research and clinical research.

Answer:  Following your advice, we added a discussion section to the revised manuscript that should be of interest to basic Scientists. There, the mechanisms of action of botulinum toxins in cancer related disorders are discussed and appropriate references are provided.  

The manuscript sent lacks a description (even a brief one, but precise and clear) of the possible mechanisms through which the effect of the toxin on the pathologies described can be explained. Some of them are obviously known, but a description of them with respect to the specific pathophysiological picture described can be useful to better frame the problem and the way to solve it. In other cases, the possible effect of the toxin from a physiological point of view is less clear. By way of example, recently, the same journal Toxins published a manuscript entitled "Mechanisms of Botulinum Toxin Type A Action on Pain" which I do not see cited, where an accurate analysis of the possible physiological mechanisms of the effect of botulinum toxin on nociceptive sensation was made.

I therefore suggest adding a part to the manuscript dedicated to the molecular and physiological mechanisms through which the toxin would act in the solution of the side effects described in the manuscript.

Answer:  Following your advice, in the added discussion section of the manuscript,  we discussed the molecular and physiological mechanisms through which the toxin alleviates different cancer related disorders ( pain and others).  The references provided for the discussion section include your suggested reference.  

Reviewer 3 Report

Comments and Suggestions for Authors

Abstract :

Line 5 : More correct to say : « By inhibiting the synaptic  release of neurotransmitters ».

Line 8 : Please do 2 sentences as it is confusing at present : issue at same level as treatment. For example : « invasion). Hence, cancer related disorders can be relieved by pyloric injection… »

Introduction :

Line 16 : « acetylcholine and pain neurotransmitters ».

Line 29 : BoNTs are also proposed as therapy against cancer development, see review : Appl Microbiol Biotechnol 2022 Jan;106(2):485-495.  doi: 10.1007/s00253-021-11741-w. Epub 2021 Dec 24. Botulinum toxin in cancer therapy-current perspectives and limitations

Tomasz Grenda 1Anna Grenda 2PaweÅ‚ Krawczyk 3Krzysztof Kwiatek 4.

It is important to mention that BoNTs can be effective in other cancer related disorders, not only the pain.

Line 37 : « provided the minimum information. »

Line 40 : « injection, results and side effects. » remove : « and ».

Results :

It would have been beneficial to focus on a wider range of cancer related disorders. The focus of this review is somehow narrow at present.

Line 49 : « gastroparesis » and « parotidectomy ».

Line 56 : cancer related pain is a critical point of this review so  would be beneficial to describe a few more cases about cancer direct pain effect since many cases are described in the literature.

Line 85 : Same for remote effect, although the example given is very interesting, it would be better to provide one or two more examples form the literature since the role of BoNT in palliative care could be established and could help patients during their terminal stage.

Line 120 : please move legend of fig 3 to the next page.

Table 2, after Line 132 : please clarify the title of column 2 as it is not clear at present.

Line 148 : please provide a table describing the results of BoNT therapy in ES cases, this would highlight the beneficial effect of the BoNT therapy in ES issue.

Line 194 : please remove spaces before : (table 3)

Line 202-203 : please clarify the sentence : « of the toxin blocking acetylcholine release from parasympathetic fibers that innervate the salivary glands ». Acetylcholine is for the parasympathic system.

Line 208 : « parotidectomy »

Conclusion :

Line 218 : Please review the first sentence : « The inhibitory effect of botulinum neurotoxins on synaptic release of acetylcholine and pain neuromediators… ». The BoNTs do not act directly agaisnt Acetylcholine, they block the release at the synaptic cleft.

Comments on the Quality of English Language

Minor English editing needed

Author Response

Abstract :

Line 5 : More correct to say : « By inhibiting the synaptic  release of neurotransmitters ».

Answer: The abstract was revised.

Line 8 : Please do 2 sentences as it is confusing at present : issue at same level as treatment. For example : « invasion). Hence, cancer related disorders can be relieved by pyloric injection… »

Answer :  Abstract was revised and expanded .  The new abstract covers your points.

Introduction :

Line 16 : « acetylcholine and pain neurotransmitters ».

Question : acetylcholine and pain neurotransmitter ( line  46-47 of marked revised manuscript)

Line 29 : BoNTs are also proposed as therapy against cancer development, see review : Appl Microbiol Biotechnol . 2022 Jan;106(2):485-495.  doi: 10.1007/s00253-021-11741-w. Epub 2021 Dec 24.Botulinum toxin in cancer therapy-current perspectives and limitations

Tomasz Grenda 1, Anna Grenda 2, PaweÅ‚ Krawczyk 3, Krzysztof Kwiatek 4.

It is important to mention that BoNTs can be effective in other cancer related disorders, not only the pain.

Answer: In the added discussion section, we have discussed your points on the possible role of botulinum toxins in cancer development and included there your cited literature.

Line 37 : « provided the minimum information. »  of marked revised manuscript

Answer: corrected in the new manuscript.   line 73.     

Line 40 : « injection, results and side effects. » remove : « and ».

Answer: Corrected in the new manuscript, line 75-76.

Results :

It would have been beneficial to focus on a wider range of cancer related disorders. The focus of this review is somehow narrow at present.

Answer: please see the revised discussion.

Line 49 : « gastroparesis » and « parotidectomy ».

Answer: the table has been corrected and the typo is corrected.

Line 56 : cancer related pain is a critical point of this review so  would be beneficial to describe a few more cases about cancer direct pain effect since many cases are described in the literature.

Answer: to our knowledge there are only two reports in the literature. Those two are reported and discussed (revised manuscript pages 3-9).

Line 85 : Same for remote effect, although the example given is very interesting, it would be better to provide one or two more examples form the literature since the role of BoNT in palliative care could be established and could help patients during their terminal stage.

 Answer: to our knowledge there are only two reports in the literature. Those two are reported and discussed (revised manuscript page).

Line 120 : please move legend of fig 3 to the next page.

Answer: done

Table 2, after Line 132 : please clarify the title of column 2 as it is not clear at present.

Answer: clarified 

Line 148 : please provide a table describing the results of BoNT therapy in ES cases, this would highlight the beneficial effect of the BoNT therapy in ES issue.

Answer: following your advice, a table was provided (page 10, revised manuscript).

Line 194 : please remove spaces before  (table 3)

Answer: Space removed

Line 202-203 : please clarify the sentence : « of the toxin blocking acetylcholine release from parasympathetic fibers that innervate the salivary glands ». Acetylcholine is for the parasympathic system.

Answer: thank you for the correction . We changed sympathetic to parasympathetic in that sentence ( Line 272-273  , revised manuscript).

Line 208 : « parotidectomy »

Answer: the word was corrected-  Thank you   Line  276  of revised manuscript

Conclusion :

Line 218 : Please review the first sentence : « The inhibitory effect of botulinum neurotoxins on synaptic release of acetylcholine and pain neuromediators… ». The BoNTs do not act directly agaisnt Acetylcholine, they block the release at the synaptic cleft.

Answer: This sentence was revised in the conclusion, line 339.

Comments on the Quality of English Language

Minor English editing needed

Answer : edition was performed

We thank you for your constructive suggestions. 

Round 2

Reviewer 2 Report

Comments and Suggestions for Authors

Accept in the present form

Author Response

Thanks.